# LM-BFS: A Framework for Landmarks in Planning

**Clemens Büchner** and **Thomas Keller**

University of Basel, Switzerland

clemens.buechner@unibas.ch, tho.keller@unibas.ch

## Abstract

It is usually prohibitively expensive to recompute landmarks in every state of a heuristic search algorithm that bases its heuristic estimate on landmarks. Successful planning systems like LAMA compute landmarks just for the initial state instead and only keep track for each state encountered during search which landmarks of the initial state are accepted or required again. The LM-A$^*$ algorithm describes a similar idea for cost-optimal heuristic search. However, it lacks support for reasonable orderings, which have recently been shown to be informative when cyclic dependencies between landmarks are exploited in the heuristic computation.

We propose the novel LM-BFS framework for best-first heuristic search with landmark-based heuristics which allows to describe a variety of algorithms in terms of five components: landmark generation, landmark graph progression, landmark graph merging, landmark graph extension, and exploration strategy. We use LM-BFS to show that considering reasonable orderings in the landmark graph progression component leads to flawed landmark graphs in LAMA, and propose a novel landmark graph extension that captures the same information flawlessly. We observe empirically that considering reasonable orderings in the extension rather than the progression component has a positive effect both for LAMA and for cost-optimal planning.

## Introduction

Algorithms based on *landmarks* have yielded state-of-the-art performance in classical planning for over a decade (Richter, Helmert, and Westphal 2008; Helmert and Domshlak 2009). Landmarks denote properties shared by every solution of a classical planning task. The literature describes several methods for generating landmarks (Zhu and Givan 2003; Hoffmann, Porteous, and Sebastia 2004; Keyder, Richter, and Helmert 2010). Some of these methods additionally produce *landmark orderings* which provide information on the order in which landmarks are reached. Orderings were originally used to decompose the planning task into several smaller sub-tasks with the landmarks as sub-goals (Porteous, Sebastia, and Hoffmann 2001). However, this approach is incomplete. An alternative approach turns landmarks into a heuristic that guides a search algorithm.

One possibility to obtain a heuristic estimate for every state encountered in search is to recompute landmarks from scratch in every state. As this is usually prohibitively expensive, successful landmark-based planning systems like the LAMA planner (Richter and Westphal 2010) compute a *landmark graph* (consisting of landmarks and landmark orderings) only once for the initial state. Whenever a state is expanded, the landmark graph is progressed by keeping track of landmarks that have been *accepted* on a path to the state and of landmarks that are guaranteed to be *required again*. The LM-count heuristic of LAMA estimates the goal distance of a state as the number of landmarks that have not been satisfied on any path leading to that state. LM-count is inadmissible and plans computed by LAMA do hence not come with any guarantee with respect to solution quality.

Karpas and Domshlak (2009) propose a closely related heuristic search algorithm that guarantees optimality. Their heuristic computes a cost partitioning among the possible achievers of landmarks and performs state space exploration with LM-A$^*$, an A$^*$ variant that keeps track of fact and action landmarks that are accepted and required again. The bookkeeping procedures of LAMA and LM-A$^*$ are mostly identical, with the exception that the latter ignores *reasonable orderings* when deciding which landmarks are accepted and required again. This is because the LAMA approach is flawed in the sense that it is possible that landmarks are not marked as accepted even though they are no longer required in every plan (and are hence no longer a landmark). While this does not affect the satisficing LAMA planner, it might render the heuristic of Karpas and Domshlak inadmissible and can hence lead to violations of the optimality guarantee of LM-A$^*$.

Büchner, Keller, and Helmert (2021) show that reasonable orderings hold valuable information that can help improve admissible heuristics. Unlike LAMA and LM-A$^*$, they embed their cyclic landmark heuristics in an algorithm that recomputes the landmark graph in every state.

In this paper, we propose a framework that not only allows to consider reasonable orderings without recomputing landmarks in every step, but also gives structure to landmark-based best-first heuristic search algorithms in general. The LM-BFS framework allows to instantiate a variety of algorithms (including LM-A$^*$ and LAMA) in terms of five components: landmark generation, landmark graph progression, landmark graph merging, landmark graph extension, and exploration strategy. We introduce these components and pro-

vide possible instantiations from related work.

Furthermore, we use the LM-BFS framework to show that considering reasonable orderings in the landmark graph progression component leads to flawed landmark graphs in LAMA. We then derive a landmark graph extension strategy that takes reasonable orderings into account and maintains correctness of the computed landmark graph. We conclude the paper with an empirical study of the discussed landmark progression and extension methods. It provides evidence that considering reasonable orderings in cost-optimal planning has a positive effect even for landmark heuristics that do not exploit cycles. Moreover, we show that our new landmark graph extension method can be substituted in LAMA in a way that we observe improvements when comparing it to the original LAMA implementation.

## Background

**Classical Planning**   We consider classical planning in the $\text{SAS}^+$ formalism (Bäckström and Nebel 1995), where a planning task is given as a 4-tuple $\Pi = \langle \mathcal{V}, s_0, G, \mathcal{A} \rangle$. $\mathcal{V}$ is a finite set of *finite domain state variables* $v$ with associated domain $\text{dom}(v)$. An atom $v \mapsto d$ is a value assignment of value $d \in \text{dom}(v)$ to $v \in \mathcal{V}$. A *partial variable assignment* is a set of atoms, each for a different variable. A *state* is a variable assignment defined on all variables $v \in \mathcal{V}$. We say $s(v) = d$ if $v \mapsto d \in s$ to denote that variable $v$ has value $d$ in state $s$. $s_0$ is the *initial state*, and the *goal* $G$ is a partial variable assignment. $\mathcal{A}$ is a finite set of *actions* $a = \langle pre, eff, cost \rangle$, where *precondition* $pre(a)$ and *effect* $eff(a)$ are partial variable assignments and $cost(a) \in \mathbb{R}_0^+$ is the *cost* of $a$.

An action $a \in \mathcal{A}$ is *applicable* in state $s$ if $pre(a) \subseteq s$. Applying an applicable $a$ in $s$ results in the state $s' = s[\![a]\!]$ where $s'(v) = d$ for all $v \mapsto d \in eff(a)$ and $s'(v) = s(v)$ otherwise. An *action sequence* $\pi = \langle a_1, \dots, a_n \rangle$ is applicable in $s = s_1$ if $s_{i+1} = s_i[\![a_i]\!]$ for all $i = 1, \dots, n$ and each action $a_i$ is applicable in $s_i$. The state that results from applying an action sequence $\pi$ in $s$ is written as $s[\![\pi]\!]$. An *s-plan* is an action sequence $\pi$ such that $G \subseteq s[\![\pi]\!]$. The *cost* of an *s-plan* $\pi = \langle a_1, \dots, a_n \rangle$ is the sum over the action costs of the sequence: $cost(\pi) = \sum_{i=1}^{n} cost(a_i)$. An *s-plan* is *optimal* if it has minimal cost among all *s-plans*.

**Landmarks, Orderings, and Landmark Graphs**   A *landmark* for a state $s$ is a propositional formula $\varphi$ over atoms that holds at some point during the execution of every *s-plan*. Hence, for every *s-plan* $\pi = \langle a_1, \dots, a_n \rangle$ there exists an $0 \leq i \leq n$ such that $s[\![\langle a_1, \dots, a_i \rangle]\!] \models \varphi$. We say that $\varphi$ is *added at time $i$* in $\pi$ iff $s[\![\langle a_1, \dots, a_i \rangle]\!] \models \varphi$ and $s[\![\langle a_1, \dots, a_{i-1} \rangle]\!] \not\models \varphi$. Furthermore, $\varphi$ is *first added at time $i$* in $\pi$ iff $s[\![\langle a_1, \dots, a_j \rangle]\!] \not\models \varphi$ for all $0 \leq j < i$. A landmark that holds in $s$ is considered to be added at time 0.

We write $i = first(\varphi, \pi)$ to denote that $\varphi$ is first added at time $i$ in $\pi$. Similarly, we write $j = last(\varphi, \pi)$ to denote the last time $j$ at which $\varphi$ is added in $\pi$. Note that $first(\varphi, \pi) = last(\varphi, \pi)$ if $\varphi$ is added exactly once in $\pi$. We say that a landmark $\varphi$ for $s$ is *required* if $last(\varphi, \pi) > 0$ for all *s-plans* $\pi$. This can be either because it does not hold in $s$ (i.e., must

---

**Algorithm 1:** The LM-BFS algorithm.

```
1  graphs[s_0] := compute_landmark_graph(s_0)
2  open.insert(s_0)
3  while open ≠ ∅ do
4      s = open.pop()
5      if G ⊆ s then return extract_plan(s);
6      G = graphs[s]
7      foreach ⟨a, s'⟩ ∈ succ(s) do
8          G' := progress_landmark_graph(G, a, s')
9          G'' := merge_landmark_graphs(graphs[s'], G')
10         graphs[s'] := extend_landmark_graph(G'', s')
11         open.insert(s')
```

be first added at time $i > 0$) or because we can infer that it must be added again at a later time.

A *landmark ordering* denotes a dependency between two landmarks. Given landmarks $\varphi$ and $\psi$ for a state $s$, there is a *natural ordering* $\varphi \rightarrow_{\text{n}} \psi$ iff $first(\varphi, \pi) < first(\psi, \pi)$ for all *s-plans* $\pi$. A natural ordering is a *greedy-necessary ordering* $\varphi \rightarrow_{\text{gn}} \psi$ if $\varphi$ is a precondition to add $\psi$ for the first time, i.e., $first(\psi, \pi) = i$ demands that $s[\![\langle a_1, \dots, a_{i-1} \rangle]\!] \models \varphi$ for $\pi = \langle a_1, \dots, a_i, \dots, a_n \rangle$. A *reasonable ordering* $\varphi \rightarrow_{\text{r}} \psi$ exists iff $first(\varphi, \pi) \leq last(\psi, \pi)$ in all *s-plans*. Note that it is possible and sometimes even necessary that $first(\psi, \pi) < first(\varphi, \pi)$ even though there is a reasonable ordering $\varphi \rightarrow_{\text{r}} \psi$ (Hoffmann, Porteous, and Sebastia 2004). In this case, $\psi$ must be added more than once.

Given a set of landmarks $\mathcal{L}$ and a set of landmark orderings $\mathcal{O}$ for state $s$, the corresponding landmark graph $\mathcal{G} = \langle \mathcal{L}, \mathcal{O} \rangle$ is a directed graph with a vertex for every landmark in $\mathcal{L}$ and an edge for every ordering in $\mathcal{O}$. An edge from node $\varphi$ to $\psi$ is labeled with the type $t \in \{\text{gn}, \text{n}, \text{r}\}$ of the corresponding ordering $\varphi \rightarrow_t \psi$ (i.e., greedy-necessary, natural, or reasonable). A *path* in $\mathcal{G}$ is a chain of edges $\pi = \varphi_1 \rightarrow_{t_1} \dots \rightarrow_{t_n} \varphi_{n+1}$ such that $\varphi_i \rightarrow_{t_i} \varphi_{i+1} \in \mathcal{O}$ for $1 \leq i \leq n$. A path is a *cycle* if $\varphi_1 = \varphi_{n+1}$.

## Landmark Framework

We propose LM-BFS, a complete framework for using landmarks in planning. It generalizes LM-A* (Karpas and Domshlak 2009) using landmarks in a generic best-first state space search. Algorithm 1 shows the pseudo code for LM-BFS. It involves five components that can be implemented in different ways:

1. computing landmarks for a given state (line 1),

2. progressing a landmark graph based on the last state transition (line 8),

3. merging landmark graphs to combine their information (line 9),

4. extending a landmark graph for the current state (line 10), and

5. exploring the state-space based on landmark heuristics (lines 4 and 11).

The literature mainly studies two of these components: on the one hand, it asks the question how to find landmarks for

a specified state (component 1). On the other hand, it asks how they can be used for planning (component 5). In the remainder of this section, we summarize some of the described methods for these two problems. There are also examples for components 2, 3, and 4 in the literature. However, the aspect to see them as interchangeable components is new and we discuss them in more detail in the next section.

**Landmark Generation** Hoffmann, Porteous, and Sebastia (2004) show that deciding whether $\varphi$ is a landmark for a state in a planning task is PSPACE-complete. Therefore, finding *all* landmarks is usually infeasible. The following list summarizes popular approximation techniques from the literature:

- Richter, Helmert, and Westphal (2008) apply an iterative back-chaining approach starting in the goal. Goal atoms are trivial landmarks. The procedure checks for every landmark which actions generate it. Common preconditions of these actions are added as new landmarks until a fix-point is reached. Additionally, Richter, Helmert, and Westphal identify landmarks in the *domain transition graph* for each single variable.

- Zhu and Givan (2003) propose a forward-propagation of landmark-labels in the *relaxed planning graph* (RPG). Atoms that hold initially are trivial landmarks, the according nodes in the RPG are labeled with themselves. Action nodes in the RPG receive all landmark labels of their preconditions, atom nodes receive the intersection of labels annotated to their achievers. All labels annotated to goal nodes of the RPG are landmarks. This technique additionally generates *action landmarks* which denote actions that must occur in every plan.

- Keyder, Richter, and Helmert (2010) build upon the method of Zhu and Givan but in the $\Pi^m$ transformation of a planning task $\Pi$. Atoms of $\Pi^m$ represent conjunctions of atoms for $\Pi$. If such a conjunction is a landmark for $\Pi^m$, then it is also a landmark for $\Pi$. This technique is unique in the sense that the landmarks can model delete effects of $\Pi$ if $m \geq 2$.

Any combination of these methods can be used to compute the landmark graph for $s_0$ in line 1 of Algorithm 1.

**Exploration Strategy** The exploration strategy component of our LM-BFS framework mostly covers two important aspects of best-first heuristic search algorithms: (i) a method that derives a heuristic estimate from a given landmark graph to insert a search node into the open list, and (ii) a method that determines which node from the open list is expanded next.

The following list contains landmark heuristics described in the literature that can be used for (i):

- Richter, Helmert, and Westphal (2008) propose the *landmark count* heuristic (LM-count) which simply takes the amount of required landmarks as heuristic estimate. LM-count is inadmissible as applying a single action can add multiple landmarks. Hence, it is not suited for cost-optimal planning.

- Karpas and Domshlak (2009) compute a *cost partitioning* over the required landmarks, where each action shares its cost among all landmarks it adds. The cost to add that landmark is the sum over all costs assigned to it by its achievers. Summing up the costs for all required landmarks is an admissible estimate of the cost to reach the goal. Bonet and Helmert (2010) show that the optimal cost partitioning is the dual view of the linear programming relaxation of the *minimum hitting set* over the possible achievers.

- Büchner, Keller, and Helmert (2021) observe that a cyclic dependency between landmarks – which can only occur in the presence of reasonable orderings – means that at least one of the landmarks in the cycle has to be achieved twice. They propose the *cyclic landmark* heuristic, which exploits this information in the form of additional constraints for the linear programming relaxation of the minimum hitting set.

Landmarks that are not required do not hold interesting information for heuristics. To avoid that the heuristic needs to decide itself which landmarks are required, landmark graphs used to compute it should ideally only contain required landmarks. Hence, we demand this property from the output of all other functions used by LM-BFS.

Typical implementations of (ii) include the A$^*$ algorithm for optimal planning as in LM-A$^*$ and greedy best-first search or weighted A$^*$ for satisficing planning (both of which are used in different phases of LAMA). LM-BFS is a special case of best-first search as it is important that path-dependent heuristics are supported. As we will see, the landmark graph of a state (and hence also the corresponding heuristic value) depends on the set of explored paths leading to that state. Stronger theoretical guarantees (e.g., with respect to optimality) can be achieved with LM-BFS if the open list is always ordered according to the available information when a node is popped, and not according to the information that was available when a node is inserted. As it is prohibitively expensive to reorder the open list in every step, we instead re-evaluate the heuristic for a state $s$ when it is popped. If the heuristic value does not change, $s$ is expanded, and it is re-inserted into the open list with its new heuristic value otherwise.

## Landmark Progression Between States

As we attempt to use landmarks for heuristic search, we need landmarks for every state encountered during search. The simplest approach to do so is by replacing lines 8–10 of Algorithm 1 with *compute_landmark_graph*$(s')$. However, Büchner (2020) finds that generating landmarks dominates the running time with this approach. Hence, this is not a feasible solution in practice.

Richter and Westphal (2010) introduce an alternative that uses the similarity of landmark graphs of similar states. More specifically, they compute exactly one landmark graph for the initial state and henceforth update it according to the observations during search. For example, if $\varphi$ is a required landmark in $s$ and action $a$ has an effect such that $\mathit{eff}(a) \models \varphi$, then $\varphi$ might not be required in $s' = s[\![a]\!]$ any-

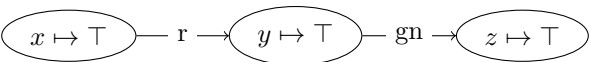

Figure 1: Example landmark graph with 3 landmarks and 2 orderings.

more. Hence, the actions or paths between two states $s$ and $s'$ can determine the differences of the according landmark graphs to some extent.

Both LM-A$^*$ (Karpas and Domshlak 2009) and the LAMA planner (Richter and Westphal 2010) use such inference techniques. They are both instantiations of LM-BFS and we explain how they implement components 2, 3, and 4 of our framework in this section. First, however, we introduce an example planning task which is later used to explain how these methods work in practice.

Consider a planning task $\Pi = \langle \mathcal{V}, \mathcal{A}, s_0, G \rangle$ where

- $\mathcal{V} = \{x, y, z\}$ with $\mathrm{dom}(v) = \{\top, \bot\}$ for all $v \in \mathcal{V}$;

- $\mathcal{A} = \{a_1, a_2, a_3\}$ such that

  - $a_1 = \langle \{y \mapsto \bot\}, \{y \mapsto \top\}, 1 \rangle$,
  - $a_2 = \langle \{y \mapsto \top, z \mapsto \bot\}, \{z \mapsto \top\}, 1 \rangle$, and
  - $a_3 = \langle \{x \mapsto \bot, z \mapsto \top\}, \{x \mapsto \top, y \mapsto \bot\}, 1 \rangle$;

- $s_0 = \{v \mapsto \bot \mid v \in \mathcal{V}\}$; and

- $G = \{v \mapsto \top \mid v \in \mathcal{V}\}$.

To simplify the notation, we write $V$ for $v \mapsto \top$ and $\neg V$ for $v \mapsto \bot$ for all $v \in \mathcal{V}$.

Figure 1 shows a landmark graph for $s_0$. Note that it is possible to identify more landmarks and orderings for this problem (e.g., $(Y \wedge Z) \rightarrow_{\mathrm{gn}} (X \wedge \neg Y)$). Nevertheless, most landmark generators are incomplete and might as well end up with the depicted graph. The landmarks can be justified because all corresponding atoms are required in the goal but do not hold in $s_0$. Hence, they must be added at some time $i > 0$ in all plans. Action $a_3$ is the only action producing $X$, but it also destroys $Y$ (i.e., it has the effect $\neg Y$). Thus, $Y$ must be last added after $X$ and hence $first(X, \pi) \leq last(X, \pi) \leq last(Y, \pi)$ which justifies the reasonable ordering $X \rightarrow_{\mathrm{r}} Y$. Action $a_2$ justifies the greedy-necessary ordering $Y \rightarrow_{\mathrm{gn}} Z$ because it is the only action producing $Z$ and it has the precondition $Y$.

The only and therefore optimal plan for $\Pi$ is $\pi = \langle a_1, a_2, a_3, a_1 \rangle$. The states (represented as tuples $\langle x, y, z \rangle$) we need to consider are

- $s_0 = \langle \bot, \bot, \bot \rangle$ as given above,

- $s_1 = s_0[\![a_1]\!] = \langle \bot, \top, \bot \rangle$,

- $s_2 = s_1[\![a_2]\!] = \langle \bot, \top, \top \rangle$,

- $s_3 = s_2[\![a_3]\!] = \langle \top, \bot, \top \rangle$, and

- $s_4 = s_3[\![a_1]\!] = \langle \top, \top, \top \rangle$.

We show that $\pi$ satisfies all ordering constraints of the landmark graph in Figure 1:

1. None of the landmarks hold in $s_0$.

2. $Y$ is first added at time 1 when applying $a_1$.

3. $Z$ is first added at time 2 when applying $a_2$; $Y \rightarrow_{\mathrm{gn}} Z$ is satisfied because $first(Y, \pi) < first(Z, \pi)$.

4. $X$ is first added at time 3 when applying $a_3$, but $s_3$ does not satisfy the goal since $a_3$ also has the effect $\neg Y$.

5. $Y$ is added (for the second and also the last time) at time 4 when applying $a_1$ (again); the goal is reached and $X \rightarrow_{\mathrm{r}} Y$ is satisfied because $first(X, \pi) \leq last(Y, \pi)$.

**LM-A$^*$** Karpas and Domshlak (2009) propose the LM-A$^*$ algorithm where each state $s$ is associated with a set of paths $\Pi$ from $s_0$ to $s$ (i.e., $s_0[\![\pi]\!] = s$ for all $\pi \in \Pi$). Given a landmark graph $\mathcal{G}_0 = \langle \mathcal{L}_0, \mathcal{O}_0 \rangle$ for the initial state, one can compute landmarks for $s$ by inference over $\Pi$. For example, if $\pi = \langle a_1, \ldots, a_n \rangle$ is such a path and $\varphi \in \mathcal{L}_0$ is not added at any time $i \leq n$, then it is a required landmark for $s$. While this argument for a single path is originally described by Richter, Helmert, and Westphal (2008), Karpas and Domshlak show that considering multiple paths is more informative.

Another way of thinking about this is the following: if all (explored) paths from $s_0$ to $s$ add a landmark $\varphi$, then $\varphi$ is probably not a required landmark for $s$. Karpas and Domshlak mark these landmarks as *accepted*. The following recursive formula collects accepted landmarks for a path $\pi = \langle a_1, \ldots, a_n \rangle$:

$$acc(\langle \rangle) = \{\varphi \in \mathcal{L}_0 \mid s_0 \models \varphi\} \tag{1}$$
$$acc(\pi) = \{\varphi \in \mathcal{L}_0 \mid s_0[\![\pi]\!] \models \varphi\} \cup acc(\pi') \tag{2}$$

where $\pi' = \langle a_1, \ldots, a_{n-1} \rangle$. Given a set of paths $\Pi$ to state $s$ (i.e., $s_0[\![\pi]\!] = s$ for all $\pi \in \Pi$), a landmark is only accepted in $s$ if it is accepted on each path individually:

$$acc(s, \Pi) = \bigcap_{\pi \in \Pi} acc(\pi). \tag{3}$$

Computing Equation 3 requires a mapping from states to sets of paths. However, the number of such paths can be exponential which renders this requirement infeasible in practice. Fortunately, the context of best-first search allows us to refine the accepted landmarks for $s$ iteratively without the need to store paths explicitly. Algorithm 2 implements LM-A$^*$ as described above in the LM-BFS framework. To do so, we need the notion of accepted landmarks incorporated into the landmark graphs. We redefine landmark graphs as triples $\mathcal{G} = \langle \mathcal{L}^+, \mathcal{L}^-, \mathcal{O} \rangle$ so that $\mathcal{L}^+$ are the accepted landmarks in the corresponding state $s$, $\mathcal{L}^-$ are required landmarks of $s$, and $\mathcal{O}$ is a set of orderings between $\mathcal{L}^+ \cup \mathcal{L}^-$. Initially, $\mathcal{L}^+_{s_0} = acc(\langle \rangle)$ and $\mathcal{L}^+_s = \mathcal{L}_0$ for all $s \neq s_0$.[1] During search, generating $s$ as a successor indicates that a new path $\pi$ to $s$ has been expanded and should be added to $\Pi$. Whenever this happens, we update $\mathcal{L}^+$ and $\mathcal{L}^-$ for the last transition in $\pi$ (lines 1–5) and merge it with findings from previous encounters with $s$ (lines 6–10). Hence, we can derive Equation 3 for each $s$ incrementally without the need to store a single path.

All algorithms discussed in this paper share the landmark graph merging strategy that is presented in Algorithm 2. We

---

[1] This is because we have no information about any $s$ except $s_0$.

**Algorithm 2:** Implementation of LM-A* in the LM-BFS framework.

**1 progress_landmark_graph**($\langle \mathcal{L}^+, \mathcal{L}^-, \mathcal{O} \rangle, a, s'$)
**2**    $accept := \{\varphi \in \mathcal{L}^- \mid s' \models \varphi\}$
**3**    $\mathcal{L}'^+ := \mathcal{L}^+ \cup accept$
**4**    $\mathcal{L}'^- := \mathcal{L}^- \setminus accept$
**5**    **return** $\langle \mathcal{L}'^+, \mathcal{L}'^-, \mathcal{O} \rangle$

**6 merge_landmark_graphs**($\langle \mathcal{L}_1^+, \mathcal{L}_1^-, \mathcal{O}_1 \rangle, \langle \mathcal{L}_2^+, \mathcal{L}_2^-, \mathcal{O}_2 \rangle$)
**7**    $\mathcal{L}^+ := \mathcal{L}_1^+ \cap \mathcal{L}_2^+$
**8**    $\mathcal{L}^- := \mathcal{L}_1^- \cup \mathcal{L}_2^-$
**9**    $\mathcal{O} := \mathcal{O}_1 \cup \mathcal{O}_2$
**10**    **return** $\langle \mathcal{L}^+, \mathcal{L}^-, \mathcal{O} \rangle$

**11 extend_landmark_graph**($\langle \mathcal{L}^+, \mathcal{L}^-, \mathcal{O} \rangle, s'$)
**12**    $\mathcal{L}_G := \{\varphi \in \mathcal{L}^+ \mid s' \not\models \varphi \text{ and } \varphi \in G\}$
**13**    $\mathcal{L}_{gn} := \{\varphi \in \mathcal{L}^+ \mid \exists \varphi \rightarrow_{gn} \psi \in \mathcal{O} \text{ and } \psi \notin \mathcal{L}^+\}$
**14**    $\mathcal{L}'^- := \mathcal{L}^- \cup \mathcal{L}_G \cup \mathcal{L}_{gn}$
**15**    **return** $\langle \mathcal{L}^+, \mathcal{L}'^-, \mathcal{O} \rangle$

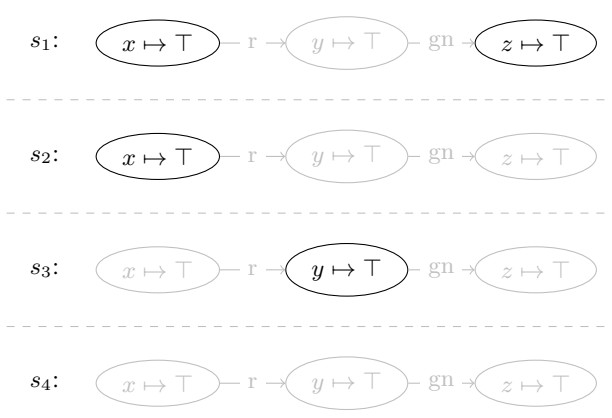

Figure 2: Evolution of the landmark graph from Figure 1 using the path-dependent landmark progression.

still consider merging an adjustable component of the LM-BFS framework as it is not hard to come up with cases where different methods could be better suited. For instance, in an LM-BFS instantiation where different landmark graphs are merged, it could pay off to remove a landmark $\varphi$ if it shares its orderings with another landmark $\psi$ and if $\psi \implies \varphi$, or to remove a reasonable ordering between two landmarks if there is also a greedy-necessary ordering between these landmarks. We do not pursue these ideas any further in this paper but plan to turn our attention to them in future work.

In the landmark graph progression and merging components, we have not taken landmarks that are *required again* into account. Richter, Helmert, and Westphal (2008) consider a landmark $\varphi$ as required again if the following criteria are satisfied: (i) $\varphi$ does not hold in $s$ and $\varphi$ must hold in the goal or (ii) $\varphi$ does not hold in $s$ and there exists an ordering $\varphi \rightarrow_{gn} \psi$ such that at least one path to $s$ does not add $\psi$. Karpas and Domshlak (2009) use the same idea in their LM-A* algorithm, and it is the only part that is still missing

in our description of an instantiation of the algorithm in the LM-BFS framework.

In the LM-BFS framework, we separate updating the landmark graph based on the last state transition (in the landmark graph progression component) from computing landmarks that are required again (in the landmark graph extension component). Lines 11–15 contain the pseudo code that adds all landmarks to $\mathcal{L}^-$ that are considered to be required again by LM-A*. Combined with an exploration strategy that corresponds to A* and a landmark generation component that uses the cost partitioning-based landmark heuristic of Karpas and Domshlak (2009), the LM-A* algorithm is fully specified as an instantiation of the LM-BFS framework.

Before we proceed, let us briefly look at how LM-A* changes the landmark graph in the example from Figure 1. In Figure 2, landmarks depicted with solid border are in $\mathcal{L}^-$ while all others are not. Note that it is not always possible to determine if a landmark is part of $\mathcal{L}^+$: landmarks with non-solid borders are know to be in $\mathcal{L}^+$, whereas landmarks with solid borders can be part of $\mathcal{L}^+$ or not. Landmarks $Y$, $Z$, and $X$ are removed from $\mathcal{L}^-$ by the landmark graph progression component in $s_1$, $s_2$, and $s_3$, respectively. Since $a_3$ deletes $Y$ which is required in the goal, $Y$ is added to $\mathcal{L}^+$ in the landmark graph extension component of $s_3$. $Y$ holds again in the goal state $s_4$, so it is again removed from $\mathcal{L}^-$ which becomes empty at this point.

**LAMA** The LAMA planner (Richter and Westphal 2010) uses a path-dependent landmark progression similar to LM-A*. In fact, the LM-A* algorithm was described subsequent to the LAMA planner. We have swapped the order in this paper because LAMA uses an additional layer of inference based on orderings. Namely, every ordering $\varphi \rightarrow_t \psi$ where $\varphi$ has not yet been added denotes that $\psi$ is required. If the ordering is natural (or stronger), it is simply impossible to add $\psi$ before $\varphi$ in any plan. If the ordering is reasonable, it means that $\varphi$ cannot be added while $\psi$ holds. Therefore, if $\psi$ holds, it must be deleted and added again after $\varphi$ (or at the same time).

Richter and Westphal incorporate this finding in the acceptance criterion of LAMA.

$$acc(\langle \rangle) = \{\varphi \in \mathcal{L}_0 \mid s_0 \models \varphi \text{ and } \not\exists(\psi \rightarrow_t \varphi) \in \mathcal{O}_0\} \quad (4)$$
$$acc(\pi) = \{\varphi \in \mathcal{L}_0 \mid s_0[\![\pi]\!] \models \varphi \text{ and} \quad (5)$$
$$\forall(\psi \rightarrow_t \varphi) \in \mathcal{O}_0 : \psi \in acc(\pi')\} \cup acc(\pi')$$

Algorithm 3 is the according implementation of the landmark progression in LM-BFS. It differs from the progress function in Algorithm 2 in line 2 where it only accepts the landmarks with no unaccepted parents. The other components (i.e., merging and extending landmark graphs) are implemented identical to LM-A*.

Applying this slightly different progression to the example of Figure 1 affects the landmark graph as shown in Figure 3. Even though $Y$ holds in $s_1$, it is not accepted since $X \notin \mathcal{L}^+$ and therefore $Y$ is not removed from $\mathcal{L}^-$. When transitioning to $s_2$, $Z$ also has an unaccepted parent (i.e., $Y \notin \mathcal{L}^+$) and is not accepted. Since $X$ has no parents, it

---

**Algorithm 3:** The LAMA progression in the LM-BFS framework.

---

1 **progress_landmark_graph**($\langle \mathcal{L}^+, \mathcal{L}^-, \mathcal{O}\rangle, a, s'$)
2    $accept := \{\varphi \in \mathcal{L}^- \mid s' \models \varphi$ **and**
                  $\forall \psi \rightarrow_t \varphi \in \mathcal{O} : \psi \in \mathcal{L}^+\}$
3    $\mathcal{L}'^+ := \mathcal{L}^+ \cup accept$
4    $\mathcal{L}'^- := \mathcal{L}^- \setminus accept$
5    **return** $\mathcal{G}' = \langle \mathcal{L}'^+, \mathcal{L}'^-, \mathcal{O}\rangle$

---

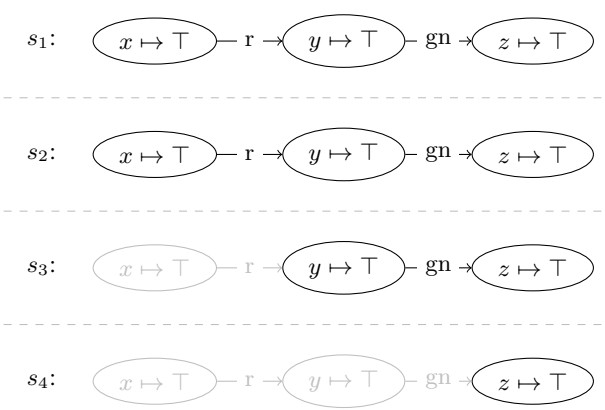

Figure 3: Evolution of the landmark graph from Figure 1 using the LAMA landmark progression.

is removed from $\mathcal{L}^-$ and added to $\mathcal{L}^+$ when added through $a_3$. Finally, $Y$ is accepted in the transition to $s_4$ as this time, $X \in \mathcal{L}^+$. We note that $Z \in \mathcal{L}^-$ for all states, even in the goal state $s_4$. Therefore, LAMA considers $Z$ to be required even though it is not when we reach the goal.

**Theorem 1.** *The* LAMA *progression is unsound.*
*Let $s$ be a state of a planning task and let $s' = s[\![a]\!]$ be the result of applying action $a$ in $s$. Furthermore, let $\mathcal{G}$ be a landmark graph for $s$ containing only required landmarks and let $\mathcal{G}'$ be the result of applying Algorithm 3 for $\mathcal{G}$, $a$, and $s'$.*

*Some nodes in $\mathcal{G}'$ might not constitute required landmarks for $s'$.*

The example outlined above proves Theorem 1. The underlying problem occurs in the transition from $s_1$ to $s_2$. It is true that the reasonable ordering justifies that $Y$ is required (i.e., $Y \in \mathcal{L}^-$) because it must be deleted to add $X$ and must then be added again to reach the goal. However, when $Z$ is added, $Y$ was already first added. Therefore, the greedy-necessary ordering $Y \rightarrow_{\text{gn}} Z$ is satisfied and $Z$ should be accepted at this point (i.e., it should be added to $\mathcal{L}^+$). LAMA makes a faulty deduction here because it assumes that landmarks that have not been accepted previously cannot have been first added. While this is indeed the case for Equations 1 and 2, it is not so for Equations 4 and 5. Therefore it is wrong to decide which orderings are not satisfied based on LAMA's accepted landmarks.

Ultimately, the LAMA progression applied to our running example yields a heuristic value of 1 in the goal when using the LM-count heuristic. Richter and Westphal point out this possibility in their planner description:

> "However, the heuristic may also assign a non-zero value to a goal state. This happens if plans are found that do not obey the reasonable orderings[2] in the landmark graph, in which case a goal state may be reached without all landmarks being accepted." (Richter and Westphal 2010, pp. 149)

We see in the example, however, that the problem can occur for arbitrary states, as $Z \in \mathcal{L}^-$ in $s_2$, $s_3$, and $s_4$.

Richter and Westphal try to avoid having such problematic orderings in their landmark graphs. For example, they do not consider reasonable orderings $\varphi \rightarrow_{\text{r}} \psi$ where $s_0 \not\models \varphi$ but $s_0 \models \psi$. This is because these orderings denote that $first(\psi, \pi) < first(\varphi, \pi)$ for all plans $\pi$ and therefore no plan obeys that ordering. Furthermore, Richter and Westphal ensure that landmark graphs are acyclic because a cycle also means that a reasonable ordering must be disobeyed (Büchner, Keller, and Helmert 2021). Our example shows, however, that these conditions are not sufficient to ensure that all plans obey the remaining reasonable orderings. Therefore, the issue persists and LAMA considers some landmarks to be required even if they are not.

Due to the assumption that landmark graphs obey all reasonable orderings, LAMA makes use of another type of landmark orderings called *obedient-reasonable orderings*. They are originally introduced by Hoffmann, Porteous, and Sebastia (2004) and are only valid if each plan obeys all reasonable orderings. Again, our example shows that this is not necessarily the case and therefore obedient-reasonable orderings are not valid. We currently see no use case for them in cost-optimal planning. Whether or not they can be used in some way or another is an open question that we leave for future work.

The take home message here is that the LAMA progression is not suitable for cost-optimal planning because it can yield unsound landmarks. Since LAMA is a satisficing planner, though, using an inadmissible heuristic is not necessarily problematic. On the contrary, LAMA has shown great success in the International Planning Competitions 2008 and 2011. However, in some cases such as our example, it suffers from the flaws in its landmark progression. Furthermore, it imposes limitations on the landmark graph which make it less informed than it could be.

**Admissible Reasonable Orders** We have seen that the reason why reasonable orders lead to false landmarks is because some landmarks that have been encountered are not marked as accepted. Essentially, all landmarks $\varphi$ from which we know that $first(\varphi, \pi)$ lies before $s$ on all known paths $\pi$ from $s_0$ to $s$ must be part of the set $\mathcal{L}^+$ of $s$, and all landmarks $\varphi$ from which we know that $last(\varphi, \pi)$ lies after $s$ on all known paths from $s$ to a goal state $s^\star$ must be part of the set $\mathcal{L}^-$ of $s$. It is the violation of this essential invariant of the LM-BFS framework that leads to the flawed landmark graph of LAMA.

---

[2]A plan $\pi$ *disobeys* a reasonable ordering $\varphi \rightarrow_{\text{r}} \psi$ if $first(\psi, \pi) < first(\varphi, \pi)$.

| **Algorithm 4:** The reasonable order landmark graph extension in the LM-BFS framework. |
| :--- |
| 1 **extend_landmark_graphs**($\langle \mathcal{L}^+, \mathcal{L}^-, \mathcal{O}\rangle, s'$) |
| 2      $\mathcal{L}_G := \{\varphi \in \mathcal{L}^+ \mid s' \not\models \varphi \text{ and } \varphi \in G\}$ |
| 3      $\mathcal{L}_{\text{gn}} := \{\varphi \in \mathcal{L}^+ \mid \exists \varphi \rightarrow_{\text{gn}} \psi \in \mathcal{O} \text{ and } \psi \notin \mathcal{L}^+\}$ |
| 4      $\mathcal{L}_{\text{r}} := \{\varphi \in \mathcal{L}^+ \mid \exists \psi \rightarrow_{\text{r}} \varphi \in \mathcal{O} \text{ and } \psi \notin \mathcal{L}^+\}$ |
| 5      $\mathcal{L}'^- := \mathcal{L}^- \cup \mathcal{L}_G \cup \mathcal{L}_{\text{gn}} \cup \mathcal{L}_{\text{r}}$ |
| 6      **return** $\langle \mathcal{L}^+, \mathcal{L}'^-, \mathcal{O}\rangle$ |

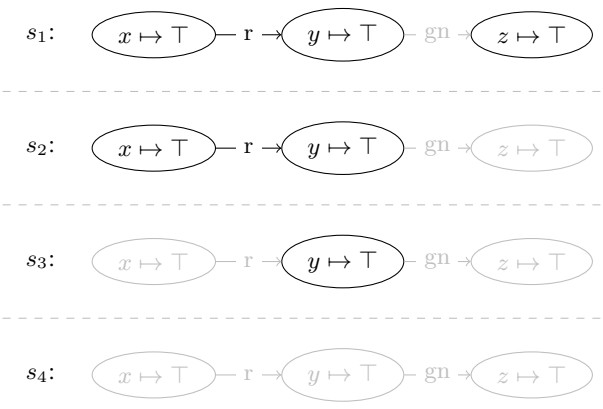

Figure 4: Evolution of the landmark graph from Figure 1 using the fixed LAMA landmark progression.

As it turns out, it is surprisingly simple to fix this: returning to the landmark graph progression component that is used in LM-A* ensures that all landmarks which have been seen on all paths to $s$ are in $\mathcal{L}^+$, and using the landmark graph extension component that is given in Algorithm 4 (which differs from Algorithm 2 only by the adding line 4 and merging the result in line 5) adds all landmarks to $\mathcal{L}^-$ from which we know that they are required again due to their open reasonable ordering.

Figure 4 shows the effect of applying this fixed version of the LAMA progression to our running example. We see that all landmarks remain in $\mathcal{L}^-$ of $s_1$ for the same reasons as in Figure 3, but unlike in the LAMA progression, $Y$ is also part of $\mathcal{L}^+$ of $s_1$. In $s_2$, we see that the flaw of the LAMA landmark progression is fixed: $Z$ is correctly removed from $\mathcal{L}^-$ in the landmark graph progression and it is not added again in the landmark graph extension since $Y$ is in $\mathcal{L}^+$. In the following two steps, the algorithm correctly removes the landmarks $X$ and $Y$ from $\mathcal{L}^-$, yielding an LM-BFS algorithm that both classifies landmarks correctly as accepted and required again and is able to deal with reasonable orderings.

## Experimental Evaluation

The Fast Downward planner (Helmert 2006) uses landmarks along the lines of LM-BFS. We implement the discussed landmark progressions in version 20.06 of Fast Downward. We refer to them as LM-A*, LAMA, and ARO for the admissible reasonable orderings. Since ARO is not limited to acyclic graphs, we also consider ARO$^c$ which keeps cycles in the landmark graph. Furthermore, LAMA$^o$ denotes the orig-

inal version of LAMA which considers obedient-reasonable orderings. Experiments are conducted on Intel Xeon-Silver 4114 processors running on 2.2 GHz with a time limit of 30 minutes and a memory limit of 3.5 GB.

**Cost-Optimal Planning** One contribution of this paper is the applicability of reasonable orderings for cost-optimal planning. We generate landmarks for the initial state using the following methods: LM$^{\text{RHW}}$ (Richter, Helmert, and Westphal 2008), LM$^{h^1}$ (Keyder, Richter, and Helmert 2010), and LM$^{\text{BJOLP}}$ (Domshlak et al. 2011). We use the optimal cost partitioning heuristic described by Karpas and Domshlak (2009) combined with an A* exploration strategy. Evaluating LAMA in this context is pointless, as it renders the heuristic inadmissible due to the unsound landmarks. Instead, we evaluate LM-A*, ARO, and ARO$^c$ on 1827 benchmark tasks from the optimal tracks of the International Planning Competitions 1998–2018.

The only difference between LM-A* and ARO is the consideration of reasonable orderings. Hence, the set of required landmarks can only become larger in ARO due to the additional landmark graph extension. Since ARO$^c$ does not discard orderings for cycles, it considers even more reasonable orderings which can lead to even more required landmarks. A super-set of a set of landmarks can only yield equal or higher heuristic estimates using the optimal cost partitioning heuristic. Therefore, ARO$^c$ dominates ARO, which in turn dominates LM-A*, for the same sets of paths to each state.

All tested landmark generators benefit from the additional information of ARO and ARO$^c$. However, the task coverage is merely unaffected with an increase between 0 and 4 from LM-A* to ARO and between 1 and 5 from LM-A* to ARO$^c$. Both ARO and ARO$^c$ have a slight advantage in terms of the average running time, but it does not seem to be systematic, as there are many instances where LM-A* is faster. Memory is never a limitation, though ARO and ARO$^c$ also perform slightly better in this category. This is most likely due to the significantly smaller number of expanded states before the last $f$-layer. Figure 5 plots the expansions for all considered landmark generators and compares LM-A* with ARO$^c$. Tasks where both methods expand the same amount of states are filtered out.

**Satisficing Planning** In a second experiment, we compare the LAMA configuration of Fast Downward with different landmark graph progressions. LAMA uses an anytime search which aims to find a (potentially sub-optimal) plan fast and improve it until the time or memory limit is reached, or it cannot further improve the solution (Richter and Westphal 2010). It uses the FF heuristic (Hoffmann and Nebel 2001) and the LM-count heuristic (Richter, Helmert, and Westphal 2008).

We evaluate the different LM-BFS instantiations considering reasonable orderings on the 2772 planning tasks from the satisficing tracks of the International Planning Competitions 1998–2018. Our baseline is the LAMA$^o$ progression as the original LAMA planner considers obedient-reasonable orderings. The reported numbers denote the average from 5

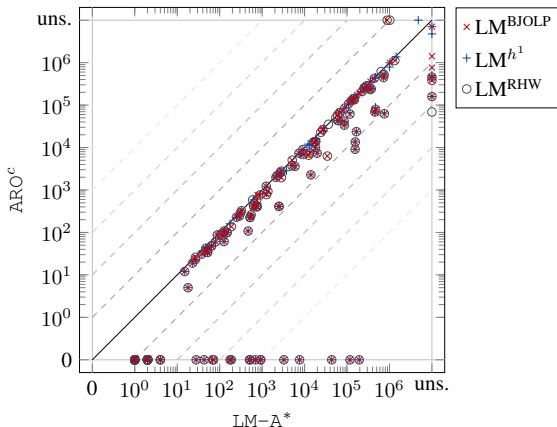

Figure 5: Number of expanded states before the last $f$-layer.

|  | $\text{LAMA}^o$ | LAMA | ARO | $\text{ARO}^c$ |
|---|---|---|---|---|
| $\text{LAMA}^o$ | – | 2.4 | 98.0 | 352.8 |
| LAMA | **5.0** | – | 99.6 | 355.0 |
| ARO | **103.4** | **101.4** | – | 269.4 |
| $\text{ARO}^c$ | **411.6** | **410.0** | **324.6** | – |

Table 1: Per-task comparison of the final plan cost. Each cell denotes how many tasks (on average) were found to have a lower cost in the row method compared to the column. The winner of each pairwise comparison is highlighted in boldface.

runs with different successor orderings to smooth out random noise effects.

Problem coverage is merely unaffected by the choice of progression strategy. While $\text{LAMA}^o$ and LAMA solve 2381 tasks on average, both ARO and $\text{ARO}^c$ solve 2379 tasks. In terms of running times, we observe a slight advantage for $\text{LAMA}^o$ and LAMA which does not seem to be systematic, though. In terms of memory and state expansions, we observe the opposite, a minor improvement of ARO and $\text{ARO}^c$. These measures all propose that there is no significant difference between the compared progression strategies.

There is, however, a difference in terms of plan quality. Table 1 provides insights concerning the plan quality, i.e., the cost of the best plan found before the planner terminates. It compares pair-wise which strategy finds a cheaper plan in how many tasks. Interestingly, LAMA without obedient-reasonable orderings performs almost the same as $\text{LAMA}^o$; the plan cost differs in only 7.4 tasks on average with a slight advantage for LAMA over $\text{LAMA}^o$. Since an analysis of obedient-reasonable orderings is out of the scope of this paper, we do not investigate this further.

There are more tasks with different plan costs when comparing to ARO and $\text{ARO}^c$. However, neither technique strictly dominates the others in this regard. Yet, we can observe a pattern, namely that $\text{ARO}^c$ consistently improves upon the plan quality with between 55 and 60 more tasks with cheaper solution costs in all comparisons. In summary, these results indicate that the admissible reasonable orderings progression can replace the LAMA progression with no major drawbacks. Moreover, it imposes no constraints on the used landmark graphs which means it benefits even more if better landmark generators for $s_0$ become available in the future.

## Conclusion

In this paper we propose the LM-BFS framework. It denotes a best-first search for planning based on landmarks and consists of five interchangeable components: (1) landmark generation, (2) landmark graph progression based on state transitions, (3) landmark graph merging, (4) landmark graph extension, and (5) exploration strategies based on landmark heuristics. The literature introduces planning systems that are instantiations of this framework: LM-A$^*$ (Karpas and Domshlak 2009) is an optimal search strategy and LAMA (Richter and Westphal 2010) is a satisficing planner.

Both of these planning techniques compute a landmark graph for the initial state and infer landmarks for all other states based on the search history. This analysis is done by deciding which landmarks are accepted to reach a state $s$ and also which landmarks are required again to reach the goal from $s$. LAMA uses one additional criteria compared to LM-A$^*$. Namely, it deduces that some landmarks are added too early in the sense that they must be added again later. However, the implementation of this idea is flawed which can lead to unsound landmarks in some cases. We propose a novel landmark graph extension component that incorporates the same idea but is guaranteed to generate sound landmarks at all times. Our empirical evaluation shows that LAMA does not suffer from this change. Furthermore, this allows for the first time to use reasonable orderings for cost-optimal planning without the need to recompute landmark graphs in every state. We provide evidence that doing so has a positive effect on planner performance.

## Acknowledgments

We thank Augusto B. Corrêa, Patrick Ferber, and Malte Helmert for their feedback during the writing process. We have received funding for this work from the European Research Council (ERC) under the European Union's Horizon 2020 research and innovation programme (grant agreement no. 817639). Moreover, this research was partially supported by TAILOR, a project funded by the EU Horizon 2020 research and innovation programme under grant agreement no. 952215.

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
