# OpenReview forum: "LM-BFS: A Framework for Landmarks in Planning"
_icaps-conference.org/ICAPS/2021/Workshop/HSDIP — HSDIP 2021_

### Official Review · AnonReviewer1 · 2021-05-26
**Good paper introducing a generalized framework for using landmarks in heuristic search.**

**Confidence:** 4
**Overall Score:** Accept

**Review:**

The paper introduces a new general framework for using landmarks in heuristic search. The authors make the following contributions:
- introduce new framework
- prove landmarks used by the LAMA planner to be unsound
- introduce a method to use reasonable orders between landmarks in an admissible heuristic

The new framework splits the overall search procedure into 5 steps:
1) computation of landmarks
2-4) updating + reasoning with landmark graphs during search based on explored states
5) search space exploration

This is modular and general enough to capture many known landmark heuristics + search strategies like LM-A* or the LAMA search.
As such, this contribution (slightly) generalizes previous attempts and provides a clean framework.

The authors show that the landsmarks used by LAMA are not sound, i.e., that the heuristic might use landmarks for a state that are in fact no longer landmarks.
The paper proposes a fix that enables the use of these landmarks to obtain an admissible heuristic. The new variant is also able to handle cyclic landmark graphs.

In particular the last two points are a bit preliminary. Given that the original LAMA landmarks were flawed, it would probably be good to formally prove that the fixed variant is indeed admissible.
The authors could also elaborate on the usage of cyclic landmark graphs.

The experimental evaluation shows that using reasonable orders and cyclic landmark graphs can be beneficial for optimal planning with A* in terms of state expansions. I'd like to see also some data regarding the per-state evaluation time. Is this significantly affected? You mention that overall the basic variant LM-A* is faster in many instances.
In satisficing planning, the authors report that coverage and runtime are mostly unaffected. There seems to be an advantage of the fixed LAMA landmarks with reasonable orderings in terms of solution quality. I wonder if this is really systematic. Yes, there seems to be a trend that solutions with ARO^c are cheaper, but do you have some evidence that this is more than just noise? In the end, the LAMA search configuration is somewhat complex and not easy to analyse. A minor change in a few heuristic values might completely change the investigated search space. I suggest to conduct some additional experiments with random successor reordering to check if the advantage of ARO^c persists.

Overall, this is a good paper, it is well-written, and the topic is clearly in the scope of HSDIP. Thus, I recommend to accept it.


Some minor things:
- second to last paragraph in BG: should it be "s-plans" instead of "plans"? (twice)
- the enumeration in the LM framework section seems to be overwide.
- description of ZG LMs: "All labels in annotated"
- description of RHW LMs: "more than one landmarks"
- last paragraph of LM framework: the sentence starting with "as we will see" is a bit weird
- Example: there is actually more than one plan in your example. The sequence a3,a1 for example can be appended arbitrarily often.
There is only one *minimal* plan.
- the action that "produces X" "destroys" Y, not "\neg Y", right?
- page 6 in the middle of the right paragraph: s_0 \not\models \phi
- how to you get to 2772 instances from satisficing IPC tracks? Shouldn't this be in the order of the optimal tracks?

---

> ### Author Response · Authors · 2021-05-28
> **Authors' Response to this Valuable Review**
>
> We thank the reviewer for the detailed comments.
>
> ### Soundness of our Landmark Update
> We agree that a formal proof about the soundness of our update is missing.
> We expect it to be straightforward after some modifications in the background and minor changes in the part about the update techniques.
> These reorganizations are already planned for preparing this paper for a conference submission.
>
> ### Cyclic Landmark Graphs
> Thank you for pointing this out.
> This should definitely be highlighted earlier in the paper, as it challenges the misconception that cyclic landmark graphs harm the planner performance.
>
> ### Per-State Evaluation Time
> We will look into this and report on it if there is something interesting to see.
> However, we do not expect a significant difference in this regard:
> the only difference are the additional reasonable orderings and it should be fast to check them for required again landmarks.
> Furthermore, we believe the per-state evaluation time to be dominated by solving the LP for the optimal cost partitioning.
>
> ### Comparison to LAMA
> Thanks for the suggestion about testing LAMA with different successor orderings.
> Indeed, this is a good point and we have not yet tested this thoroughly.
>
> Our main goal of this experiment was to show that LAMA does not suffer from replacing its landmark update.
> Since our alternative update with reasonable orderings fixes the flaws of the LAMA update, we still believe it is worthwhile to consider it even if it does not improve the overall solution quality or search time.
>
> ### Minor Comments
> Thank you for catching and reporting these issues, we will address all of them before submitting the CRC.
>
> The size of our IPC satisficing benchmark set is as used in the Fast Downward issue experiments.
> There exists the DEFAULT_SATISFICING_SUITE in common_setup.py which we used for this purpose.
> Its counterpart is the DEFAULT_OPTIMAL_SUITE and their respective sizes correspond to the ones reported in our paper.

---

> > ### Comment · AnonReviewer1 · 2021-06-01
> > **Thank you!**
> >
> > Thank you very much for the detailed response and clarifications!

---

### Official Review · AnonReviewer2 · 2021-05-27
**Interesting paper focusing on a framework generalizing LAMA and LM-A***

**Confidence:** 4
**Overall Score:** Accept

**Review:**

The paper proposes a framework for heuristic search with landmarks. It shows that the proposed framework generalizes LAMA and LM-A*, and it shows how to extend LM-A* with landmarks with "reasonable ordering".

The paper is easy and interesting read. There isn't much theory-wise, but it provides enough practical insights to be accepted in HSDIP, in my opinion.

Minor issue:
Eq. (5): missing "and" between "\psi \in acc(\pi')" and "\forall ..."

---

> ### Author Response · Authors · 2021-05-28
> **Thanks**
>
> We thank the reviewer for the positive feedback.
>
> We will make sure to fix the typo for the CRC, thanks for the pointer.

---

### Decision · Program_Chairs · 2021-06-10

**Decision:**

Accept

**Comment:**

Congratulations, all reviewers agree that the paper is a clear accept.